| Editor's Pick | Computational Biology | Research Article

# A hidden confounder for microbiome studies: medications used years before sample collection

Oliver Aasmets,[1] Nele Taba,[1] Kertu Liis Krigul,[1] Reidar Andreson,[1,2] Estonian Biobank Research Team,[1] Elin Org[1]

**ABSTRACT** Medication usage is a known contributor to the inter-individual variability of the gut microbiome. However, medications are often used repeatedly and for long periods, a notion yet unaccounted for in microbiome studies. Recently, we and others showed that not only the usage of antibiotics and antidepressants at sampling, but also past consumption, is associated with the gut microbiome. This effect can be "additive"—the more a medication is used, the stronger the impact on the microbiome. Here, by utilizing retrospective medication usage data from the electronic health records and the observational Estonian microbiome cohort shotgun metagenomics data set ($n$ = 2,509), we systematically evaluate the long-term effects of antibiotics and human-targeted medications on the gut microbiome. We show that past usage of medications is associated with the gut microbiome. For example, the effects of antibiotics, psycholeptics, antidepressants, proton pump inhibitors, and beta-blockers are detectable several years after use. Furthermore, by analyzing a subcohort ($n$ = 328) with a second microbiome characterization, we show that similar changes in the gut microbiome occur after treatment initiation or discontinuation, possibly indicating causal effects.

**IMPORTANCE** This is the first study using detailed retrospective medication usage data from electronic health records to systematically assess the long-term effects of medication usage on the gut microbiome. We identified carryover and additive effects on the gut microbiome for a range of antibiotics and non-antibiotic medications, such as benzodiazepine derivatives, antidepressants and glucocorticoids, among others. These findings highlight a collateral effect of diverse drug classes on the gut microbiome, which warrants accounting for long-term medication usage history when assessing disease-microbiome associations.

**KEYWORDS** gut microbiome, electronic health registries, drugs, human-targeted drugs, antibiotics, antidepressants, psycholeptics, beta-blockers, benzodiazepines, long-term drug effects, metagenomics, drug usage, drug prescriptions, medications

The human gut microbiome is acknowledged as an important contributor to our well-being and is considered a therapeutic target for health interventions. The structure and composition of this complex ecosystem reflect our health status, consumption of drugs, dietary choices, lifestyle, and the environment we live in (1–5). As a result, a significant proportion of research is focusing on figuring out how we can use this information for disease diagnostics (6, 7), identifying disease risks (8, 9), and personalization of drug usage (10, 11). However, recent evidence shows that past exposures long preceding the sample collection can affect the gut microbiome, a direction less studied.

We and others have recently shown that the usage of antibiotics in the past (>6 months ago) can affect the microbiome composition independent of the antibiotic usage within 6 months of sample collection (1, 12–14). Moreover, this effect can be

Address correspondence to Oliver Aasmets, oliver.aasmets@ut.ee, or Elin Org, elin.org@ut.ee.

The authors declare no conflict of interest.

See the funding table on p. 15.

"additive"—the more drugs used in the past, the stronger the effect on the microbiome composition (1, 12, 13). Importantly, we also showed in a mouse model how this long-term effect of antibiotic usage can disrupt mucus function, including mucus growth and penetrability, and may increase abdominal fat weight (15). In addition to antibiotics, there are implications for the long-term effects of antidepressants (1) and beta-blockers (16), and it has been shown that the longer duration of proton pump inhibitors (PPIs) consumption is associated with microbiome diversity in infants (17). Therefore, the long-term effect of drug exposure may have a major influence on our physiology, highlighting the need to understand the full extent of such effects across diverse drug classes. However, to date, a systematic evaluation of long-term drug effects on the fecal microbiome has not been carried out.

Here, we take advantage of linking the Estonian microbiome cohort (EstMB) gut metagenomics data ($n$ = 2,509) with the retrospective drug usage data from the electronic health records, allowing us to evaluate the long-term effects of drug usage systematically. We analyze the presence of long-term drug effects over a wide range of drug classes, including both human-targeted drugs (i.e., therapeutic targets of human origin) and antibiotics (i.e., therapeutic targets of bacterial origin), and assess whether the effects can be additive, as seen for antibiotic usage. Additionally, including a second time-point metagenomics data from an EstMB subcohort of 328 individuals allows us to further validate the long-term effects and study drug initiation effects.

## RESULTS

### Drug usage in the Estonian microbiome cohort

The Estonian microbiome cohort (EstMB) is a population-based volunteer cohort currently comprising 2,509 subjects (age range: 23–89 years, mean: 50.1 ± 14.93 years) who have provided blood, buccal swabs, and stool samples. In this study, we focus on the gut microbiome characterized using shotgun metagenomics sequencing. As part of the nationwide Estonian Biobank (EstBB), EstMB is supported by links to various electronic health records (EHR), and participants have provided data regarding their lifestyle and dietary preferences via questionnaires. A detailed overview of the EstMB cohort and available data is discussed in Aasmets and Krigul et al. (1). Most importantly, EHR allows us to characterize the participants' drug usage at the time of microbiome sampling and analyze the history of drug usage retrospectively in great detail (Fig. 1a). Additionally, a subcohort of the EstMB ($n$ = 328) has provided a second stool sample after a median follow-up period of 4.4 years. Hereon, we refer to the first time point of the microbiome sampling as T1 and the second time point as T2 (Fig. 1a).

At T1, 433 prescription drugs at the ATC5 level (Anatomical Therapeutic Chemical classification system) were used by the participants, representing 225 different chemical drug subgroups at the ATC4 level and 126 pharmacological subgroups at the ATC3 level (Fig. 1b). Moreover, during the 5 years preceding T1, a total of 507 different medications at the ATC5, i.e., chemical substance level, were used (251 at the ATC4 level and 138 at the ATC3 level), highlighting the wide spectrum of drug usage in the population. At T1, 857 subjects (34.2%) did not use any prescription drugs, while those taking drugs used, on average, around three different medications from diverse drug classes at the time of microbiome sampling (ATC3 mean = 2.84; ATC4 mean = 2.97; ATC5 mean = 3.02) (Fig. 1c). In the downstream analyses, we focused on drugs that were used by at least 20 subjects at T1, resulting in 56 drugs at the ATC3 level, 63 drugs at the ATC4 level, and 67 drugs at the ATC5 level. As an example, drugs at the ATC4 level with the most users at T1 include beta-blocking agents ($n$ = 234, 9.3%), proton pump inhibitors ($n$ = 211, 8.4%), and benzodiazepine derivatives ($n$ = 177, 7.1%) (Fig. 1d; Table S1), which were all often used in combination with other drugs (Fig. S1). Medications with the highest number of users over the 5 years include drugs that are commonly prescribed pro re nata (PRN; i.e., as needed), such as proton pump inhibitors (A02BC), antibiotics (J01), and psycholeptics (N05) (Fig. 1d). For these drugs, a considerable number of subjects last used the drug years ago, allowing us to study long-term drug effects (Fig. 1d).

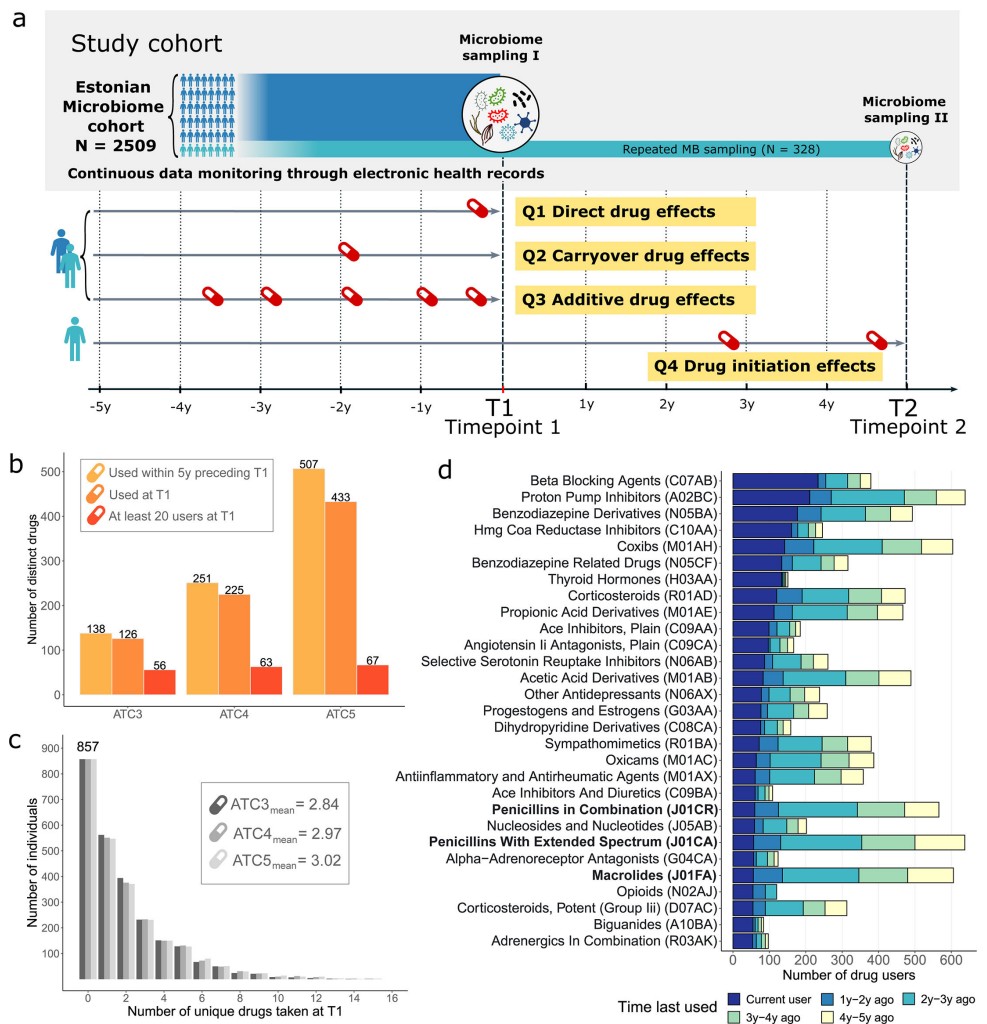

**FIG 1** Drug usage in the EstMB cohort. (a) Overview of the EstMB cohort and phenotype data available through the electronic health records (EHR). We aimed to analyze the direct effects of drug usage at the time of sampling (Q1), carryover effects induced by past drug usage (Q2), and additive effects arising from the differing amounts of drugs used in the past (Q3). In a subcohort of the EstMB (*n* = 328) that provided an additional stool sample (T2), we analyzed the alterations in the microbiome attributable to treatment initiation (Q4). (b) Total number of different drugs used in the EstMB at different ATC levels. (c) Distribution of the number of drugs used by the participants at T1 across different ATC levels. (d) The number of drug users at T1 on ATC4 level; drugs with at least 50 users at T1 are shown. Drugs are ordered according to the number of active drug users. Antibiotics are highlighted in bold. ATC, Anatomical Therapeutic Chemical classification; Q, study question.

By taking advantage of the EHR, we aimed to analyze the direct effects of drug usage at the time of sampling (Q1), drug carryover effects induced by past drug usage (Q2), additive effects arising from the different amounts of drug used in the past (Q3), and validate the cross-sectional findings by analyzing the effects of drug initiation and discontinuation between the two time points (Q4) (Fig. 1a).

## Effects of active drug usage on the gut microbiome composition (Q1)

Firstly, to characterize the effect of active drug usage, we aimed to identify drugs and drug classes associated with the gut microbiome at T1 via comparing individuals taking a drug at T1 with those who had not taken the corresponding drug during the 5 years preceding T1. When analyzing human-targeted drugs, we excluded individuals who had used antibiotics within the 90 days preceding T1. We used linear models to assess associations with the abundance of each mOTU and logistic regression models to assess

associations with the presence-absence of each mOTU. Out of the 186 drugs assessed, 167 (89.8%) were associated with either alpha diversity, beta diversity, or the abundance of at least one bacterial species (FDR ≤ 0.1) (Fig. 2; Tables S2 and S3). The drug effect on the beta diversity was evident in 96 of the 186 drugs (PERMANOVA on Aitchison distance; FDR ≤ 0.1; Fig. 2a; Table S2), indicating that drugs, including human-targeted medications, can significantly alter the microbiome on the compositional scale. For example, at the chemical subgroup level (ATC4), drugs explaining the most inter-individual variability included beta-blockers (PERMANOVA, $R^2$ = 0.149%, FDR = 0.0010), macrolides (PERMANOVA, $R^2$ = 0.142%, FDR = 0.0010), and biguanides (PERMANOVA, $R^2$ = 0.116%, FDR = 0.0010) (Fig. 2a; Table S2). Most of these drug classes, including beta-blockers, macrolides, and benzodiazepine derivatives, were negatively correlated with alpha diversity metrics, especially with observed richness (Fig. 2b). Additionally, we observed that the more unique drugs used at the time of sampling, the lower the microbial alpha diversity (Fig. S2a). Altogether, these observations are concordant with previous research and support the notion of drug usage as a significant factor in explaining inter-individual microbiome variability (5, 12, 18, 19).

We further focused on the associations between drug usage and the presence-absence (PA) and abundance (centered log-ratio [CLR] transformed) of 530 mOTUs (20), which were prevalent in at least 10% of the cohort. To eliminate the potential carryover effects, we compared the active drug users with the subjects who had not taken the drug of interest during the 5 years preceding T1. All analyses were adjusted for body mass index (BMI), gender, and age. To account for the potential confounding by diseases, lifestyle, and usage of other drugs, we conducted a thorough analysis, as described by Forslund et al. (12). Interestingly, we identified only a few drug-microbe associations that were confounded by the presence of a disease, and despite a few exceptions, we did not find significant confounding by other drugs, although such results have been shown *in vitro* (21) (Table S3). In all downstream analyses, we focused on confidently deconfounded associations. Interestingly, we found the effect directions for microbial taxa to be highly similar across drug classes, indicating a common signal of drug usage (Fig. 2c; Table S3). This observation was evident in both abundance and presence-absence analyses. For example, *[Clostridium] asparagiforme/lavalense* [06317], *[Clostridium] clostridioforme/bolteae* [03442], and *[Clostridium] citroniae* [04828] from the family *Lachnospiraceae* are positively correlated with the usage of beta-blocking agents, macrolides, biguanides, and PPIs, among other drugs (Fig. 2c; Table S3). As human-targeted drugs have been shown to inhibit bacterial concentrations similarly to antibiotics *in vitro* (21–23), we next examined whether a similar observation can be made *in vivo*. For that, we built machine learning models on the CLR-transformed microbiome data to predict the usage of various antibiotic subclasses and tested their ability to predict the usage of other drug classes in unobserved data. Indeed, we observed that, for example, the model aimed to detect the usage of macrolides (AUC = 0.94 for macrolides) could identify the usage of biguanides (AUC = 0.71) and selective serotonin reuptake inhibitors (AUC = 0.67), and the model aimed to detect penicillin usage (AUC = 0.75 for penicillin) can, among others, detect the usage of antidepressants (AUC >0.58) and various corticosteroids (Fig. S2b; Table S4). In contrast, we found several bacterial species, such as *Dorea longicatena* [03693] and *Eubacterium* species [12260], that are significantly associated with antibiotics, but not with host-targeted drugs (Table S3). Thus, at least partially, the effect of antibiotics and human-targeted drugs overlaps.

We then compared our results to the previously published population studies (5, 12) and the most comprehensive *in vitro* study (22). We observed a moderate overlap with the results from population studies we could compare, especially with antibiotic and PPI usage (Table S5). For example, PPIs are positively associated with the abundance of oral microbes *Streptococcus parasanguinis* and *Veillonella parvula* in our study, as well as in Vila et al. (19), Nagata et al. (5), and in Forslund et al. (12). More specifically, PPIs esomeprazole and omeprazole are positively associated with *Streptococcus parasanguinis*, and esomeprazole is additionally associated with *Veillonella parvula*. However, there were

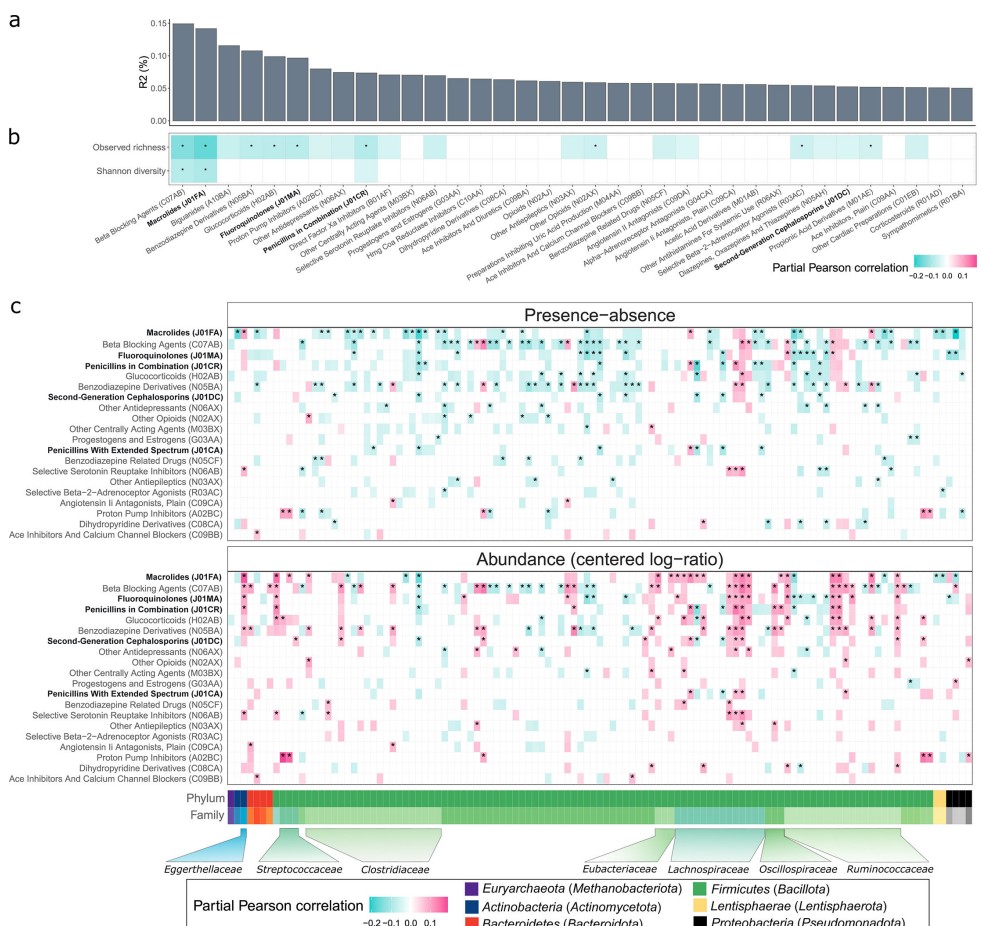

**FIG 2** Active drug usage effects on the microbiome. (a) The inter-individual variance of the microbiome explained by individual drug classes. (b) Partial Pearson correlation between drug classes at the ATC4 level and observed richness and Shannon diversity index. (c) Partial Pearson correlation between drug classes at the ATC4 level and bacterial species analyzed as presence-absence and as abundance (centered log-ratio transformed) data. The results of both analyses refer to the same bacteria, i.e., the same column refers to the same bacterial species in presence-absence and in abundance analyses. Further, only drugs and microbes with at least 10 confidently deconfounded nominally significant associations ($P < 0.05$) in two analyses combined are shown, while drugs are presented in descending order based on the number of associations detected (top 20 drugs shown). The colored cells indicate nominally significant and confidently deconfounded associations ($P < 0.05$). *FDR $\leq$ 0.1 and confidently deconfounded. Antibiotics are highlighted in bold.

also inconsistencies. Among others, for example, we observed positive associations between benzodiazepine derivatives and *Dorea formicigenerans* and *Ruminococcus torques*, whereas Nagata et al. (5) observed significant negative effects for the same comparisons. Therefore, although identifying robust signals *in vivo* remains a challenge, large population-based cohorts can provide valuable insights.

We next assessed whether the drugs belonging to the same pharmacological subgroup act similarly in terms of the effect on the microbiome. We observed several notable examples with varying effects. For example, beta-blockers metoprolol (PERMANOVA, $R^2 = 0.104\%$; linear/logistic regression, number of univariate hits $n_{univariate} = 104$) and nebivolol (PERMANOVA, $R^2 = 0.069\%$; linear/logistic regression, $n_{univariate} = 18$) display remarkable differences, indicating that drugs used for a similar condition and belonging to the same pharmacological and chemical subgroup can have a different impact on the fecal microbiome. A similar discrepancy can be seen for benzodiazepine derivatives, such as diazepam vs. alprazolam, and PPIs, such as omeprazole vs. pantoprazole/esomeprazole (Tables S2 and S3). Drug dosage is a likely factor that can complicate dissecting the microbe-drug associations and be a reason for the differences in effect

size for drugs belonging to the same drug group. For example, we found that the effects of PPI omeprazole on some microbes can be observed only for higher doses (Fig. S2c; Table S6). Therefore, well-powered studies that are fit to analyze the effects of distinct drugs at the chemical substance level (i.e., ATC5) and that include dosage information are likely to further elucidate the drug-microbe associations.

## Long-term carryover effects on the microbiome composition are independent of recent drug usage (Q2)

Next, we focused on the drug-taxa pairs for which we identified associations with active drug usage in Q1 (univariate linear or logistic regression, FDR ≤ 0.1) and analyzed whether these effects were observable when the drugs were last used years before the microbiome sampling. This serves the aim to detect the long-term carryover effects that are present even if the drug has not been recently used. For that, we compared the former (>1 year before T1) drug users with the subjects who had not used the drug in the 5 years preceding T1 (Fig. 3a; Fig. S3a; Table S7). To detect the carryover effect and its possible duration, we used similar models to those in Q1, but with four different scenarios comparing the drug non-users with individuals who had last used the drug >1 year, >2 years, >3 years, or >4 years ago.

Indeed, we observed potential carryover effects for various drug classes (78/186, 41.9% of drugs). The effects of past usage at the ATC4 level were most clearly evident for different antibiotic subclasses but also for human-targeted drugs, such as benzodiazepine derivatives, biguanides, proton pump inhibitors, and antidepressants. As with active drug usage (Fig. 2c), the carryover effects with microbes were similar across the drug classes (Table S7). Remarkably, drug-microbe associations for several broad-spectrum antibiotics, such as macrolides and penicillins in combination, as well as for human-targeted drug classes, e.g., benzodiazepine derivatives and antidepressants, could be identified even if they were last used more than 3 years before the microbiome sampling (Fig. 3a). Considering the carryover effects of antibiotics, we tried to pinpoint the duration of the effect of antibiotic usage on microbiome richness. The diversity of antibiotic users does not seem to reach the observed richness of the antibiotic non-users, irrespective of the antibiotic load and time from the last antibiotic treatment (Fig. 3b).

## Long-term drug effects are partly of additive nature (Q3)

The detected carryover effects raised a follow-up question: can the long-term effects also accumulate, as previously shown for antibiotics (1, 12, 13)? Specifically, does the previous drug usage or the amount of it explain additional variability in the microbiome on top of the active drug usage? To answer this question, we fit and compared three competing models (M1, M2, and M3) with increasing complexity: M1 includes only active drug usage, M2 includes active and past usage (within the 5 years preceding T1) as a binary trait, and M3 includes active usage and amount of past usage measured by the number of prescriptions bought out during the 5 years preceding T1. We observed that for a majority of the drugs, the model including past usage (M2) was best fitting for at least some of the drug-microbe associations, supporting the presence of long-term drug effects (Fig. 3c; Fig. S3b; Table S8). Moreover, the model including the amount of drug used (M3) was best fitting for several drugs, including human-targeted drugs, such as beta-blockers, benzodiazepine derivatives, and glucocorticoids. For example, benzodiazepine derivatives show an additive effect on *Eisenbergiella tayi* [03446] and several *Clostridiales* species (Table S8). Of note, the number of prescriptions for human-targeted drugs with continuous treatment regimes is closely related to drug adherence—the more prescriptions taken in the past, the better adherence to drug intake. Further, to characterize the effect of active and long-term drug usage on the overall variability of the fecal microbiome, we carried out a multivariate analysis of the explained variance. This analysis confirmed that long-term drug usage has a significant effect on microbiome variability, independent of active drug usage (Fig. 3d; Table S9). Moreover, the long-term effects exceed the effects of active drug usage in terms of variance explained (0.74% and

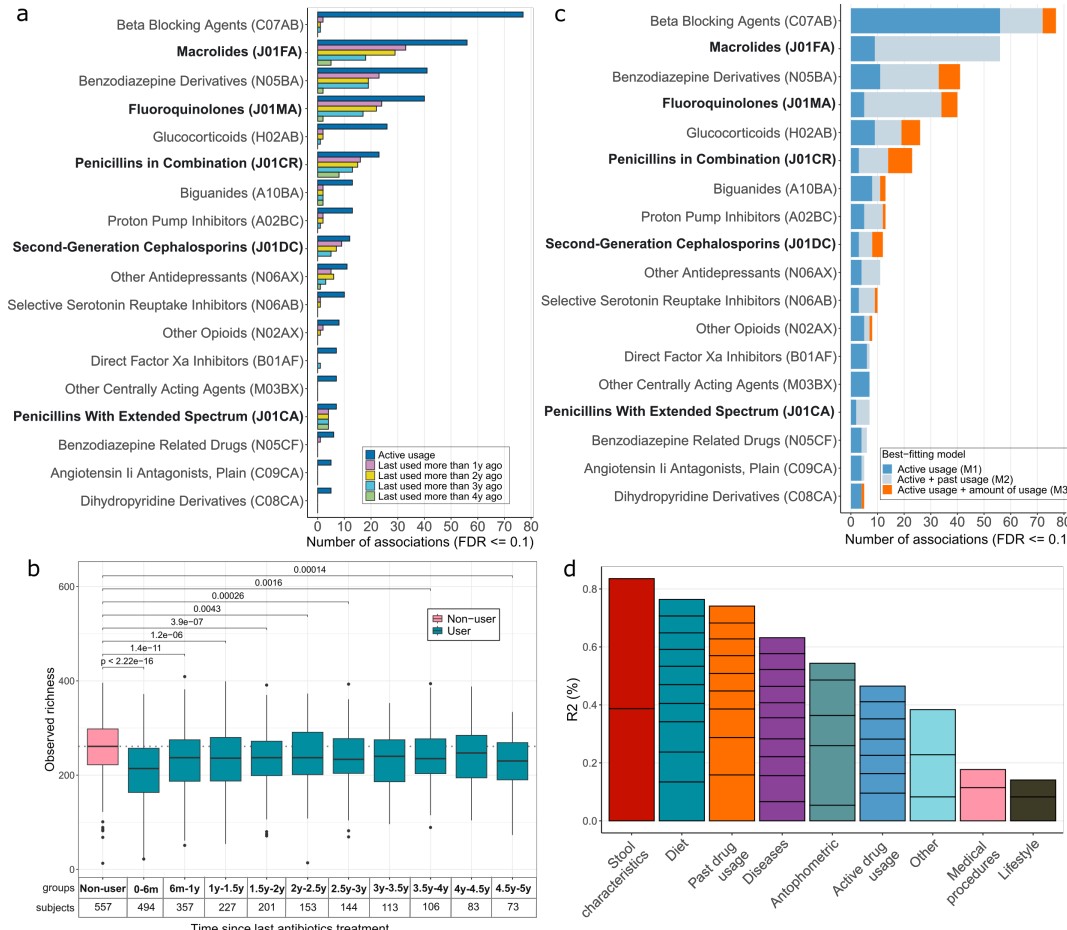

**FIG 3** Long-term drug effects on the gut microbiome. (a) Drug carryover effects. The number of univariate associations between subjects not taking the drug and subjects having taken the drug more than 1 to 4 years prior to the microbiome sample collection (CLR-transformed abundance). (b) Long-term effects of antibiotic use on the microbiome. (c) Additive drug effects. The proportion of the univariate associations identified with active drug usage according to the model that best describes the association (CLR-transformed abundance). In addition to active drug usage as a binary trait, active drug usage together with past drug usage as a binary trait or the amount of drug usage as a continuous trait is considered. (d) Variance explained in the gut microbiome composition according to different factor groups. The bars indicating the variance explained by factor groups are further divided into individual factors according to their contribution (individual factors are listed in Table S9). Past drug usage indicates drugs used during the 5 years preceding sample collection at T1. Antibiotics are highlighted in bold.

0.47%, respectively). Importantly, we observe that such long-term drug usage effects can also confound disease-microbe associations, highlighting the importance of accounting for past drug usage (Fig. S4).

## Microbiome measurements from the two time points confirm long-term drug effects (Q4)

Next, to assess the effect of drug initiation or discontinuation on the microbiome, we focused on the microbiome changes between T1 and T2 (median follow-up period 4.4 years) in a subcohort with the microbiome measured at two time points (*N* = 328). First, we compared the individuals who initiated drug usage between T1 and T2 and were active users at T2 with the drug-naive participants (Tables S10 and S11). Neither the controls nor the drug initiators used the drugs 5 years before T1. Despite a limited sample size, we identified changes in the microbiome attributable to the use of drugs at T2 for broad-spectrum antibiotics, such as penicillins and macrolides, and human-targeted drugs, including proton pump inhibitors, benzodiazepine derivatives, and glucocorticoids (Fig. 4; Table S11). We also observed drug initiation effects for drugs

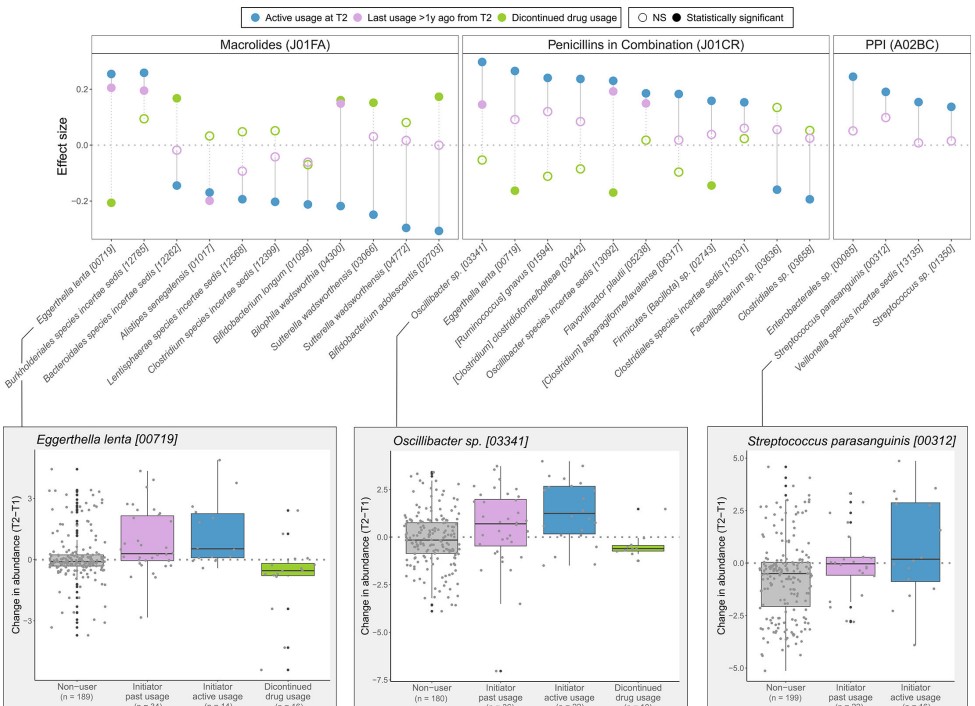

**FIG 4** Drug initiation and discontinuation effects. Associations between drug initiation, discontinuation, and changes in the microbiome between T2 and T1 are shown for macrolides, penicillins, and PPIs. Dark blue refers to associations with active drug usage, light blue refers to associations with drug usage that happened more than 1 year before T2, and green refers to associations with discontinued drug use. Effect size refers to partial Pearson correlation adjusted for BMI, gender, and age.

with no effects identified for active usage in the cross-sectional setting (Q1) (Table S11). However, the proportion of nominal hits in the Q4 analysis was significantly higher in the group that also had an association with active usage in Q1 (20.5% vs 6.4%), indicating that Q1 and Q4 analyses capture similar signals. Similarly, to study carryover effects, we compared drug initiators, who last used the drug more than 1 year before T2, with the drug-naive subjects. This allowed us to verify the presence of carryover effects for macrolides, penicillins, and human-targeted drugs, such as PPIs and selective serotonin reuptake inhibitors (Fig. 4; Table S11). All of the significant associations shown in Fig. 4 are also seen in cross-sectional analyses (T1) assessing active usage (Q1) and carryover effects (Q2), with one exception: the carryover effect of macrolides (J01FA) on *Bilophila wadsworthia* [04300] was not identified. Further, among penicillins, both the active and past usage of amoxicillin and clavulanic acid (J02CR02) were significantly associated with changes (T2-T1) in the abundance of *Flavonifractor plautii* [05238], *Oscillibacter* sp. [03341], and *Oscillibacter* species *incertae sedis* [13092]—all these associations were also observable in Q1 and Q2. Thus, analyzing two time points provides a means for identifying robust drug-microbiome associations. As a last step, we analyzed the drug discontinuation between the two time points. The observed effects with opposite direction compared to drug initiation further support our findings (Fig. 4; Table S11).

## DISCUSSION

Here, we present the first systematic evaluation of long-term drug usage effects on the fecal microbiome across various drug classes, including a large number of human-targeted drugs. While the research so far has mostly been focusing on the effects of drug use at the time of microbiome sampling, our unique data set allowed us to demonstrate that the effect on the microbiome can still be seen years after the last usage of a drug. After thoroughly accounting for confounding, we observed that out

of the 186 drugs analyzed, 167 (89.8%) are associated with the microbiome, while 78 of them (46.7%) display long-term effects. In addition to antibiotics, e.g., macrolides, fluoroquinolones, and different penicillin classes, several human-targeted drugs, such as beta-blockers, benzodiazepine derivatives, glucocorticoids, PPIs, biguanides, and antidepressants, display effects on the microbiome observable years after previous drug intake. Furthermore, this effect can be additive, i.e., it depends on the amount of drug used in the past (measured by the number of prescriptions for a drug). By using second time point samples from 328 individuals, we verified the long-term effects of antibiotics, such as penicillins in combination and macrolides, as well as human-targeted drugs, such as PPIs and selective serotonin reuptake inhibitors, displaying a likely causal effect. Drug usage has been previously shown to explain a significant proportion of inter-individual variability in the fecal microbiome composition (3, 5, 18). Moreover, the effect of drug usage has been shown to supersede the effect of disease, further highlighting the significant impact of drug usage (5, 12). Our results extend this observation, demonstrating that past drug usage can explain additional variability independent of active drug usage, suggesting that the effect of drug usage has been underestimated. Importantly, not only antibiotics but also human-targeted drugs are among the drugs explaining additional variability. Considering that the drug burden can be highly variable across populations, and specific drugs, e.g., antidepressants, antibiotics, and benzodiazepine derivatives, may exhibit varying prescription patterns across different countries (24–26), drug usage can possibly lead to differences in the microbiome composition between the populations. For example, it has been shown that per capita antibiotic use correlates with the prevalence of antimicrobial resistance genes (27). As antibiotic resistance mechanisms have also been shown to protect against human-targeted drugs (22), this concurrently may affect the abundance of specific microbes in different populations and result in an overall shift in the microbiome composition. Also, the drug burden is usually higher in older age groups, so it can be expected that the microbiome in the older populations is even more affected. Similarly, clinical cohorts might also have a higher drug burden compared to the general population cohorts from volunteer-based biobanks, where healthier individuals are usually recruited. Thus, the long-term effects and varying drug burden can induce differences between cohorts and could consequently confound cross-cohort comparisons when unaccounted for.

Our findings are in agreement with the current notion that several human-targeted drugs seem to act similarly to antibiotics (22). The associations between human-targeted drugs and the presence-absence of the bacteria suggest that drug usage is associated with the probability of observing bacteria in the sample. Given that most of the associations were negative, human-targeted drugs indeed seem to lower the diversity by eliminating specific taxa. The same holds for the long-term effects of human-targeted drugs. This can be one of the mechanisms that result in the long-term effects we are observing. The long-term effects leave a question of whether the human-targeted drugs might also lead to detrimental microbiota-mediated physiological effects, as has been shown with the history of antibiotic use (15). Given that human-targeted drugs are often taken continuously throughout life, not for short periods, which is the case for antibiotics, the physiological effects can be even more profound. Moreover, participants usually consume several different classes of drugs, and only a few individuals did not take any prescription medications in the 5-year observational period. Such drug-drug interactions may also be additive, i.e., long-term consumption of multiple drugs with a similar scope can supersede the detrimental effect of both individual drugs, but this remains to be studied in larger samples. Further, we identified that several disease-microbiome associations can be confounded by long-term drug usage. Therefore, disentangling disease-drug effects can be further improved by accounting for long-term drug usage in addition to active drug usage (28) and other host variables (29).

Surprisingly, our results showed that benzodiazepine derivatives have an even broader effect on the whole microbiome composition as well as on the presence and abundance of individual microbial species compared to several antibiotic classes.

Moreover, benzodiazepine derivatives show remarkable carryover effects several years after their use, which is comparable to effects observed with the broad-spectrum antibiotic classes. The drugs belonging to benzodiazepine derivatives are well-known anti-anxiety medications that are often misused and have a high potential for drug abuse (29). Concernedly, the use of these drugs has increased over time (30–32). In addition, we observed that at the chemical substance level, alprazolam (ATC5 N05BA12, e.g., Xanax) and diazepam (ATC5 N05BA01, e.g., Valium) affect the microbiome on different scales, with alprazolam showing a broader impact on the microbiome. This was also previously observed by Nagata et al. (5). Taking into account the rising popularity of benzodiazepines, the noted difference in the effects on the microbiome by alprazolam versus diazepam might be a valuable input for future therapy decisions and warrants further investigation. Further, the same notion could hold for other drugs, where drugs assigned for the same health condition can have an unequal magnitude of effects on the microbiome, and consequently, choosing the drug with less long-term harm on the microbiome might be favored.

*In vitro* studies have indicated that more than 100 antimicrobials and over 200 human-targeted drugs can inhibit the growth of gut commensals in isolation, which, however, might be attenuated in a community (21–23). Nevertheless, to date, imitating the complexity of the gut microbiome and the intestinal environment remains a limitation for *in vitro* studies. Therefore, population-based cohorts using metagenomics sequencing data in combination with detailed drug usage characterization are essential to understanding the drug effects on the microbiome. One of the biggest strengths of our study is the possibility of using electronic health records (EHR) to analyze drug intake. When compared to self-reported medication data, EHR does not suffer from underreporting the use of some drugs and allows to characterize long-term drug usage. The Estonian microbiome cohort is a population-based volunteer cohort which allows us to study drug usage in a general population. Thus, when compared to the disease cohorts and clinical cohorts, the drug burden, general health, and polypharmacy are likely to have a smaller impact. Additionally, the comprehensive statistical approach taken was supplemented by the measurements from a second time point, which provided an internal validation of the results. We observed that more than 500 different ATC5-level drugs have been used within the last 5 years in our sample ($N$ = 2,509). However, the sample size for analyzing the effects of all of these drugs on the gut microbiome is limited. Still, comparing the results from our study to previous findings has allowed us to pinpoint several robust drug-microbe interactions, such as the case of PPI usage and the increase in the oral microbes, like *S. parasanguinis* or *Veillonella parvula*. Some limitations of our study should be noted when interpreting the results. Our study focuses only on prescription-based drugs, and thus, the long-term effects of over-the-counter drugs remain to be studied. Also, we assume that when a prescription is bought out, subjects also take the drug. Similar to other volunteer-based biobanks, our cohort suffers from gender imbalance—there are more females than males, and the cohort participants are likely to be more interested in their health. Additionally, the results might be influenced by the use of relative abundance data as opposed to absolute abundances, which might not always ideally reflect biologically relevant changes in the microbiome.

In conclusion, our results demonstrate that medications, not limited to antibiotics, have a long-term effect on the fecal microbiome. Further, we highlight the importance of accounting for the history of drug usage when assessing disease-microbiome associations. Taken together, our results expand the understanding of drug effects on the microbiome, and we encourage researchers to focus on the long-term drug effects whenever feasible.

## MATERIALS AND METHODS

### Estonian microbiome cohort and metadata preprocessing

The Estonian microbiome cohort (EstMB) was established in 2017 when stool, oral, and blood samples were collected from 2,509 Estonian Biobank (EstBB) participants. In the EstMB cohort, 70.3% are women ($n$ = 1764), and 29.7% are men ($n$ = 745) (age range: 23–89 years old, mean: 50.1 ± 14.93 years old; body mass index range: 15.1–54.0 kg/m$^2$, mean: 26.5 ± 5.34 kg/m$^2$). The majority of the participants are of Estonian origin (98.4%). A detailed overview of the EstMB, including omics and phenotypic data availability, is described in Aasmets and Krigul et al. (1). Out of the 2,509 participants, 328 provided an additional stool sample after a median follow-up period of 4.4 years. We refer to the first time point of the microbiome sampling as T1 and the second time point as T2. All participants of the EstMB cohort gave informed consent for the data and samples to be used for scientific purposes.

Here, we rely on the electronic health records (EHR) data, which is available to all the EstBB participants, including the EstMB participants. The EHR data on the diseases and medications were obtained from the Estonian Health Insurance Fund, the Estonian Cancer Registry, and the two biggest hospitals in Estonia (University of Tartu Clinic and North Estonia Medical Centre). The Anatomical Therapeutic Chemical (ATC) classification system was used to define drug classes, and drugs with at least 20 active users during the first microbiome sampling point were included. For antibiotics and anti-infectives, active usage was defined as usage within 90 days before the sample collection. For human-targeted drugs, we assessed the active drug usage according to the amount and time of purchase. This resulted in 56 drugs at the ATC3 level, 63 drugs at the ATC4 level, and 67 drugs at the ATC5 level for downstream analysis. When analyzing additive drug effects, the number of prescriptions for past usage is based on drugs purchased during the 5 years preceding the microbiome sampling, whereby the most recent prescriptions that indicate active usage are excluded from the count. We assume that when a prescription has been bought out, subjects have also consumed the drug. The summary of the number of drug users at T1 and the average number of prescriptions in 5 years per drug are summarized in Table S1.

### Microbiome sample collection and DNA extraction

The participants collected a fresh stool sample immediately after defecation using a sterile Pasteur pipette and placed it into a polypropylene 15 mL conical tube. At the first time point (collected between 2017 and 2019), the participants brought the samples to the study center themselves (1). They were instructed to time their sample collection as close as possible to the visiting time in the study center and keep the samples in the fridge (+4°C) until transportation. The median transport time between sampling and arrival at the freezer in the core facility was 3 hours and 25 minutes (mean: 4 hours and 34 minutes; minimum: 3 minutes; and maximum: 51 hours and 15 minutes [note: additionally, three outlying measurements were between 96 and 290 hours]). The transport time was not significantly associated with alpha (Spearman correlation, $P$-value 0.949 for observed richness and 0.464 for Shannon index) nor beta diversity ($P$-value 0.061, $R^2$ 0.0005). In the second time point, the participants were instructed to send the samples by post immediately after taking the sample and time their sample collection to avoid shipping on the weekends. The median time between sampling and arrival at the freezer in the core facility for the second time point was 53 hours and 1 minute (mean: 59 hours and 22 minutes; minimum: 34 minutes; maximum: 168 hours and 7 minutes [note: additionally, one outlying measurement was 508 hours and 27 minutes]). The transport time was not significantly associated with alpha (Spearman correlation, $P$-value 0.159 for observed richness and 0.580 for Shannon index), but was associated with beta diversity ($P$-value 0.0088, $R^2$ 0.0047). Upon arrival, all samples were stored at −80°C until DNA extraction. Microbial DNA extraction was performed using the QIAamp DNA Stool Mini Kit (Qiagen, Germany). Around 200 mg of stool was used as a starting material

following the DNA extraction kit manufacturer's instructions for the extraction. DNA was quantified from all samples using a Qubit 2.0 Fluorometer with dsDNA Assay Kit (Thermo Fisher Scientific). NEBNext Ultra DNA Library Prep Kit for Illumina (NEB, USA) was used for generating sequencing libraries following the manufacturer's recommendations. Briefly, 1 μg DNA per sample was used as input material. Index codes were added to attribute sequences to each sample. The DNA sample was fragmented by sonication to an average size of 350 bp. DNA fragments were end-polished, A-tailed, and ligated with the full-length adaptor for Illumina sequencing with further PCR amplification. Finally, PCR products were purified (AMPure XP system), and libraries were analyzed for size distribution using the Agilent2100 Bioanalyzer and quantified using real-time PCR.

## Metagenomics data analyses

The shotgun metagenomics paired-end sequencing was performed by Novogene Bioinformatics Technology Co., Ltd., using Illumina NovaSeq6000 platform, resulting in $4.62 \pm 0.44$ Gb of data per sample (insert size 350 bp, read length $2 \times 250$ bp). First, the reads were trimmed for quality and adapter sequences. The host reads that aligned to the human genome were removed using *SOAP2.21* (parameters: -s 135 L 30 v 7 m $200 \times 400$) (33). The taxonomic profiling was done using the *mOTUs2.5* tool with default parameters (20). In total, 14,213 marker gene-based operational taxonomic units (mOTUs) were identified. Alpha and beta diversity analyses were carried out on the whole identified composition. For univariate analysis, mOTUs, which were detected in at least 10% of the samples, were used to limit the number of tests carried out, resulting in 530 mOTUs. Filtered mOTU profiles were also used as predictors for building classification models. We did not rarefy the counts to avoid loss of data.

## Statistical analysis

All statistical analyses were done using the R (v. 4.0.1) software.

## Diversity analysis

We used observed richness and the Shannon diversity index to assess alpha diversity. Shannon index was calculated using the vegan package (v2.5-6) (34). Associations between observed richness, Shannon index, and drug usage were analyzed as described in the univariate analysis section. To calculate the between-sample distances for beta diversity analysis, we used the Euclidean distance on the centered log-ratio (CLR) transformed microbiome species-level profile (35). We tested the associations between drug usage and microbiome composition with permutational analysis of variance (PERMANOVA) (36) on the between-sample distances using 10,000 permutations for the *P*-value calculation (Table S2). To carry out PERMANOVA, we used the adonis function from the vegan package. To apply the CLR transformation, zero counts were imputed with a pseudocount equal to half of the minimal non-zero relative abundance value.

## Univariate analysis

To analyze active drug usage effects (Q1 in Fig. 1a), we compared subjects taking a drug at T1 with subjects who had not taken the drug during the 5 years preceding T1. To analyze the effect of human-targeted drugs, we excluded subjects who had used antibiotics within the 90 days preceding T1. To associate the abundance of each mOTU with drug usage, we used linear models adjusted for age, BMI, and gender:

$$\mathrm{CLR}(\mathrm{taxa}_i) \sim \mathrm{age} + \mathrm{gender} + \mathrm{BMI} + \mathrm{I}(\mathrm{drug\ usage}), \quad i = 1\ldots530 \tag{1}$$

We used the same linear regression models for observed richness and Shannon diversity index. We report the partial Pearson correlation coefficient that is adjusted for age, BMI, and gender as the effect size. Alternatively, to associate the presence-absence

(PA) of each mOTU as a binary trait with the drug usage, we used logistic regression models adjusted for age, BMI, and gender:

$$\text{logit}(\text{PA}(\text{taxa}_i)) \sim \text{age} + \text{gender} + \text{BMI} + I(\text{drug usage}), \quad i = 1...530 \qquad (2)$$

where we considered relative abundance value >0 to indicate the presence of a taxon. We accounted for multiple testing using the Benjamini–Hochberg procedure.

To analyze the carryover effects (Q2 in Fig. 1a), we compared subjects who had last taken the drug more than $x$ years ($x = 1, 2, 3, 4$) before T1 with those who had not taken the drug during the 5 years preceding T1. Similarly to active drug usage, linear and logistic regression models, as in formulas 1 and 2, were used for the analysis. For carryover analysis and other downstream analyses, we compared the microbe-drug pairs where the effect of active drug usage was identified (analysis Q1, FDR ≤ 0.1) (Table S7).

To analyze additive drug effects (Q3 in Fig. 1a), we compared the fit of three competing models:

$$\text{M1: } \text{CLR}(\text{taxa}) \sim \text{age} + \text{gender} + \text{BMI} + I(\text{active drug usage}) \qquad (3)$$
$$\text{M2: } \text{CLR}(\text{taxa}) \sim \text{age} + \text{gender} + \text{BMI} + I(\text{active drug usage}) + I(\text{past drug usage}) \qquad (4)$$
$$\text{M3: } \text{CLR}(\text{taxa}) \sim \text{age} + \text{gender} + \text{BMI} + I(\text{active drug usage}) + \#\text{prescriptions} \qquad (5)$$

Models are increasing in complexity: model M1 indicates only the effect of active drug usage, M2 indicates the effect of active and past usage, and M3 indicates the effect of active drug usage and the amount of drug usage during the past 5 years as indicated by the number of prescriptions bought out in the past (independent of active usage). Akaike information criteria (AIC) were used to compare the fit of the models. A more complex model was considered to be a better fit if the difference in AIC with the simpler model was greater than 2 (Table S8). Additionally, we performed the same analysis with presence-absence data, where in M1, M2, and M3, instead of CLR(taxa) is logit(PA(taxa)).

To study the drug initiation effects (Q4 in Fig. 1a), we compared the subjects initiating the drug usage between T1 and T2 with the subjects who did not use the drug between the time points or at T2. Additionally, neither the controls nor the drug initiators used the drugs 5 years before T1. We further divided the drug initiators into drug initiators who used drugs at T2 and initiators who last used the drug more than a year before T2 to study the active drug usage effects and carryover drug effects. Drugs with at least 10 initiators or discontinuers were analyzed (Table S10). We used linear models adjusted for age, BMI, and gender to analyze drug initiation-related changes in the abundances of each mOTU between T1 and T2 (Table S11):

$$(\text{CLR}_{T2}(\text{taxa}_i) - \text{CLR}_{T1}(\text{taxa}_i)) \sim \text{age} + \text{gender} + \text{BMI} + I(\text{drug initiation}) \qquad (6)$$

Additionally, we compared subjects who were using the drug at T1 but did not use the drug between T2 and T1 with the subjects who had not used the drug at all, to study the effects of drug discontinuation (Q4 in Fig. 1a).

To analyze the effect of different drug dosages, we used linear models adjusted for age, BMI, and gender (Table S6):

$$\text{CLR}(\text{taxa}_i) \sim \text{age} + \text{gender} + \text{BMI} + \text{factor}(\text{drug dosage}), \quad i = 1...530 \qquad (7)$$

A likelihood ratio test was carried out to test the significance of drug dosage in the model.

## Deconfounding analysis

We carried out a rigorous *post hoc* analysis for Q1 and Q2 to identify potential confounding factors for the microbe-drug associations as described by Forslund et al. (12). Firstly, we identified naive associations between taxa and drugs, as described in the previous

section. Next, for each covariate considered as a potential confounder, we fit a pair of nested linear models adjusted for BMI, age, gender, and covariate to assess whether (i) the predictive ability of the drug exceeds the predictive ability of the covariate or (ii) the predictive ability of the covariate exceeds the predictive ability of the drug. The association identified in the first step was considered confounded when, for at least one covariate, the covariate's predictive ability exceeded the drug's predictive ability, but the opposite was not true. All prevalent diseases, all other drugs, lifestyle, anthropometric, and dietary factors described in Aasmets and Krigul et al. (1) were considered in the analysis as potential confounding factors (Table S12). The deconfounding analysis was similarly applied to identify confounders for disease-microbiome interactions. For that, the number of prescriptions for all of the drugs was considered as potential confounders in addition to the aforementioned factors to identify disease-microbiome interactions that are confounded by past drug usage. Naive disease-drug associations were identified using the linear regression model adjusted for gender, BMI, and age at sampling.

## Prediction analysis for antibiotic usage

We fit regularized linear models to predict, based on the CLR-transformed microbiome data, the active usage of antibiotic subclasses (at ATC4 level with at least 50 users at T1 [Table S1]). We implemented the elastic net models in R using the tidymodels (v0.1.1) and glmnet (v3.0-2) packages. First, we split the data in a 75:25 ratio to the training and test data sets. The models for each antibiotic subclass were tuned on the training data using repeated cross-validation (fivefold cross-validation repeated 4 times) and grid search with 50 hyperparameter combinations. The initial data split and cross-validation splits were stratified by drug usage to address class imbalance. Next, we evaluated whether the models built to detect antibiotic usage could identify the usage of other human-targeted drugs. For that, we assessed the models' ability to predict the usage of non-antibiotic drugs in the test data set, using the area under the receiver operating characteristic (AUROC) (Table S4). For the evaluation, we excluded the antibiotic users from the test set. The model building and evaluation were repeated five times on random training-test splits, and the performance estimates were averaged.

## Multivariate analysis of variance components

We used a distance-based redundancy analysis to evaluate the amount of explained variance by each factor and factor group. As factors, we considered prevalent diseases, anthropometric, lifestyle, and dietary factors, as described in Aasmets and Krigul et al. (1), along with active drug usage as binary traits and past drug usage, indicated by the number of prescriptions bought out in the past (independent of active usage) as continuous traits. We fit the initial model by combining all the factors using the dbrda function from the vegan package, followed by a forward-selection model fitting procedure using the ordistep function. For each factor in the selected model, we assessed how much it can explain the community variation, accounting for all other selected variables (Fig. 3d; Table S9). Euclidean distance on the centered log-ratio (CLR) transformed microbiome species-level profile was used to calculate between-sample distances.

## ACKNOWLEDGMENTS

The authors would like to thank Mari-Liis Tammesoo, Marili Palover, Anu Reigo, Neeme Tõnisson, Liis Leitsalu, Triinu Temberg, and Esta Pintsaar for participating in the sample collection process. We thank Steven Smit, Rita Kreevan, and Martin Tootsi for the DNA extraction process. Data analysis was carried out in part in the High-Performance Computing Center of University of Tartu.

This work was funded by the Estonian Research Council grant PRG1414 (to E.O.), the EMBO Installation grant 3573 (to E.O.), and the Biocodex Microbiota Foundation research

grant (to E.O.). E.O. was additionally supported by Estonian Center of Genomics/Road-map II project No. 16-0125.

The Estonian Biobank Research Team collected the health records data for the EstBB. E.O. and O.A. designed, and E.O. supervised the study. O.A. performed the statistical analysis. R.A. performed the bioinformatic analysis of shotgun metagenomics sequencing data. O.A., N.T., and K.L.K. interpreted the data and prepared the figures. O.A., N.T., K.L.K., R.A., and E.O. wrote the manuscript. All authors read and approved the final manuscript.

The Estonian Biobank Research Team includes Mait Metspalu, Andres Metspalu, Lili Milani, and Tõnu Esko, Institute of Genomics, Estonian Genome Centre, University of Tartu, Estonia.

## AUTHOR AFFILIATIONS

[1]Institute of Genomics, Estonian Genome Centre, University of Tartu, Tartu, Estonia
[2]Institute of Molecular and Cell Biology, University of Tartu, Tartu, Estonia

## AUTHOR ORCIDs

Oliver Aasmets http://orcid.org/0009-0001-9872-6031
Nele Taba http://orcid.org/0000-0003-1953-2819
Kertu Liis Krigul http://orcid.org/0000-0002-4195-7357
Elin Org http://orcid.org/0000-0001-8688-9717

## FUNDING

| Funder | Grant(s) | Author(s) |
| --- | --- | --- |
| Eesti Teadusagentuur | PRG1414 | Elin Org |
| European Molecular Biology Organization | Installation grant 3573 | Elin Org |
| Biocodex Microbiota Foundation | | Elin Org |
| Estonian Center of Genomics/Roadmap II | 16-0125 | Elin Org |

## AUTHOR CONTRIBUTIONS

Oliver Aasmets, Conceptualization, Formal analysis, Methodology, Software, Visualization, Writing – original draft, Writing – review and editing | Nele Taba, Formal analysis, Investigation, Methodology, Project administration, Validation, Visualization, Writing – original draft, Writing – review and editing | Kertu Liis Krigul, Investigation, Project administration, Validation, Visualization, Writing – original draft, Writing – review and editing | Reidar Andreson, Methodology, Software, Writing – original draft, Writing – review and editing | Elin Org, Conceptualization, Funding acquisition, Project administration, Resources, Supervision, Writing – original draft, Writing – review and editing.

## DATA AVAILABILITY

The metagenomic data generated in this study have been deposited in the European Genome-Phenome Archive database (https://www.ebi.ac.uk/ega/, data set accession codes EGAD00001008448 and EGAD50000001686; study accession codes EGAS00001005900 and EGAS50000001181) and are available upon request. The phenotype data include sensitive information from electronic health registers. Access to pseudomized data from the Estonian Biobank (EstBB) is granted in accordance with the Estonian Human Genes Research Act, and data access occures via the University of Tartu's secure analysis environment (SAPU server). Detailed instructions for submitting an application to access the Estonian Biobank data are available at https://genomics.ut.ee/en/content/estonian-biobank (see tab "Data Access" for detailed steps). In brief, the process starts with a preliminary inquiry to releases@ut.ee, followed by

a research application to the Scientific Advisory Committee (SAC) for scientific evaluation. The source code for the analyses is available at https://github.com/oliveraas-mets17/EstMB_drugUsage.

## ETHICS APPROVAL

The study was approved by the Ethics Committee of the University of Tartu (No 266-T10) and by the Estonian Committee on Bioethics and Human Research (Estonian Ministry of Social Affairs) (No 1.1-12/2768).

## ADDITIONAL FILES

The following material is available online.

### Supplemental Material

**Supplemental Figures (mSystems00541-25-s0001.docx).** Figures S1 to S4.
**Checklist (mSystems00541-25-s0002.xlsx).** STORMS checklist.
**Supplemental Tables (mSystems00541-25-s0003.xlsx).** Tables S1 to S12.

### Open Peer Review

**PEER REVIEW HISTORY (review-history.pdf).** An accounting of the reviewer comments and feedback.

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
