## [Reviewer comments · mSystems]

A hidden confounder for microbiome studies: medications used years before sample collection

Oliver Aasmets, Nele Taba, Kertu Krigul, Reidar Andreson, and Elin Org

Corresponding Author(s): Oliver Aasmets, Tartu Ulikool

Review Timeline:

Submission Date:	April 11, 2025
Editorial Decision:	May 19, 2025
Revision Received:	June 17, 2025
Accepted:	June 17, 2025

Editor: Kiran Patil

Reviewer(s): The reviewers have opted to remain anonymous.

Transaction Report:

DOI: <https://doi.org/10.1128/msystems.00541-25>

Re: mSystems00541-25 (A hidden confounder for microbiome studies: medications used years before sample collection)

Dear Dr. Oliver Aasmets:

Please address the remaining comments from reviewer #1. Also, please ensure that all data/code is publicly available.

Revision Guidelines

Sincerely,
Kiran Patil
Editor
mSystems

Reviewer #1 (Comments for the Author):

The revised paper of Aasmets and colleagues has improved. Nevertheless some remarks and adaptations would further ameliorate it as suggested below.

Results:

Line 153: as the oral-gut transfer has not been assessed, the association between higher prevalence of specific bacteria

commonly found in the oral cavity and PPI should not be phrased as a direct causal effect.

Line 162-163: metoprolol is excreted 95% via the kidneys while metabolites of nebivolol are excreted 48% via feces. As such they can be expected to have different impact on the gut microbiome which should be at least mentioned in the manuscript.

Data on the availability of over the counter drugs in Estonia would be helpful to interpret the data presented.

Methods

A non-significant trend on T1 for transport time on beta diversity was found but no information on the time range is available.

How was stool consistency assessed? (Extended Table 12 contains several typo's)

Extended data figures:

Please add the total users per drug (ATC4-level) in Figure 1.

The a-b-c font size of Extended data figure 2 can further be reduced.

Part of the text in Extended fig 4 is lacking.

Reviewer #2 (Comments for the Author):

The authors have satisfactorily addressed all comments. I have no further remarks.

Reviewer #1 (Comments for the Author):

The revised paper of Aasmets and colleagues has improved. Nevertheless some remarks and adaptations would further ameliorate it as suggested below.

Results

- 1) Line 153: as the oral-gut transfer has not been assessed, the association between higher prevalence of specific bacteria commonly found in the oral cavity and PPI should not be phrased as a direct causal effect.

Thank you for this suggestion. We have now modified the text and it reads as follows: "For example, PPIs are positively associated with the abundance of oral microbes *Streptococcus parasanguinis* and *Veillonella parvula* in our study, in Vila *et al.*¹⁹, Nagata *et al.*⁵ and in Forslund *et al.*¹²"

- 2) Line 162-163: metoprolol is excreted 95% via the kidneys while metabolites of nebivolol are excreted 48% via feces. As such they can be expected to have different impact on the gut microbiome which should be at least mentioned in the manuscript.

Thank you for this suggestion. We agree that pharmacokinetic principles are a likely reason for the differences in the impact on the microbiome. However, we think that the differences in the effect are relevant, independent of the excretion. For example, we also observed a similar discrepancy for benzodiazepine derivatives, diazepam vs alprazolam, which are both mostly secreted in urine. Although relevant, we think that discussion about these principles is beyond the scope of this manuscript.

- 3) Data on the availability of over the counter drugs in Estonia would be helpful to interpret the data presented.

We thank the reviewer for the question and we acknowledge that over the counter drugs can also have an effect on the microbiome, which we did not assess in the current study (as stated in discussion section within the limitations (lines 335-336, "Our study focuses only on prescription-based drugs, and thus, the long-term effects of over-the-counter drugs remain to be studied"). However, since the current paper focuses on analyzing the effect of medications that have usage data available from the electronic health records (i.e. prescription drugs), a comprehensive overview of the over-the-counter drugs is beyond the scope of this manuscript.

A full list of all the available human medicinal products in Estonia (both over-the-counter as well as prescription-based) is available to download online from the "Register of Medicinal Products" in the following link: <https://www.ravimiregister.ee/en/publichomepage.aspx?pv=PublicDownloads>.

Methods

- 4) A non-significant trend on T1 for transport time on beta diversity was found, but no information on the time range is available.

We have stated the transport times as follows for T1 (lines 373-374): *“The median transport time between sampling and arrival at the freezer in the core facility was 3 h 25 min (mean 4 h 34 min)”*, and for T2 (lines 377-378): *“The median time between sampling and arrival at the freezer in the core facility for the second time point was 53h 1 min (mean 59h 22 min).”*

We have now additionally included minimum and maximum transportation times for T1 and T2. The sentences now read as: *“In T1, the median transport time between sampling and arrival at the freezer in the core facility was 3 h 25 min (mean 4 h 34 min, minimum 3 min, maximum 51 hours and 15 min (note: additionally, three outlying measurements were between 96-290 hours)).”* for T1, and *“The median time between sampling and arrival at the freezer in the core facility for the second time point was 53h 1 min (mean 59h 22 min, minimum 34 min, maximum 168h 7 min (note: additionally, one outlying measurement was 508 hours 27 min)).”* for T2.

- 5) How was stool consistency assessed? (Extended Table 12 contains several typo's)

We thank the reviewer for this question. The stool consistency was evaluated by the participants using the Bristol Stool Scale and recategorized into three classes. We have now fixed the typos and explained the assessment for stool consistency in the Extended Table 12 variable descriptions as follows: (Constipation - Bristol stool scale 1,2; Normal - Bristol stool scale 3,4,5; Diarrhoea - Bristol stool scale 6,7).

Extended data figures

- 6) Please add the total users per drug (ATC4-level) in Figure 1.

Figure 1D reports the number of total users per drug (ATC4-level). The length of the bar in the stacked bar-plot reports the total number of users during 5 years preceding the follow-up, including the current users (numbers indicated on the x-axis), whereas each bar is divided into blocks that describe the number of individuals that consume the corresponding drug (on ATC4-level) currently or consumed last time 1-2y, 2-3y, 3-4y or 4-5y ago. The number of total users per drug is also shown in Extended Table 1.

- 7) The a-b-c font size of Extended data figure 2 can further be reduced.

We thank the reviewer for this comment. We have now reduced the a-b-c font sizes for extended data figures 2 and 3. Further, to enhance the readability of the extended data figure 3, we highlighted the antibiotics in bold and matched the color scale of extended data figure 3 with the main text figure 3.

- 8) Part of the text in Extended fig 4 is lacking.

We thank the reviewer for this comment. We have now increased the plot margins to fit all the text in the figure. We have additionally renamed some x-axis labels to be more descriptive (mental_health_status_ok -> Self-reported mental health status; health_status_ok -> Self-reported physical health status).

Re: mSystems00541-25R1 (A hidden confounder for microbiome studies: medications used years before sample collection)

Dear Dr. Oliver Aasmets:

Your manuscript has been accepted, and I am forwarding it to the ASM production staff for publication. Your paper will first be checked to make sure all elements meet the technical requirements. ASM staff will contact you if anything needs to be revised before copyediting and production can begin. Otherwise, you will be notified when your proofs are ready to be viewed.

Sincerely,
Kiran Patil
Editor
mSystems